# Exploring the Relationship between Forest Structure and Health

**Jinki Kim** [1,*] 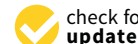**, Duk-Byeong Park** [2] **and Jung Il Seo** [3,*]

[1]  Department of Landscape Architecture, Kongju National University, 182 Shinkwan-dong, Gongju 32588, Chungcheongnam-do, Korea

[2]  Department of Regional Development, Kongju National University, 182 Shinkwan-dong, Gongju 32588, Chungcheongnam-do, Korea; parkdb84@kongju.ac.kr

[3]  Department of Forest Resources, Kongju National University, 182 Shinkwan-dong, Gongju 32588, Chungcheongnam-do, Korea

*  Correspondence: jkkim12@kongju.ac.kr (J.K.); jungil.seo@kongju.ac.kr (J.I.S.); Tel.: +82-41-330-1447 (J.K.); +82-41-330-1302 (J.I.S.)

**Abstract:** There is abundant evidence that green space in urban neighborhood is associated with physical activity and it is well known that physical activity contributes to human health. Physical activity fosters normal growth and development, can reduce the risk of chronic diseases, and can make people feel better and function better. Evidences also show that exposure to natural places can lead to positive mental health outcomes, whether a view of nature from a window, being within natural places, or exercising in these environments. The study aims to identify the factors of forest structure and socioeconomic characteristics influencing adults' physical activity and health. A sample of 148,754 respondents from the Korea Community Health Survey, conducted in 2016, was analyzed. Measures included frequency of physical activity, stress, depression, and landscape metrics of forest patch. Hierarchical multiple regression analysis, controlling for socio-demographic characteristics, revealed that larger forest patches and the more irregular shapes were associated with more physical activity. The study also showed that the shape of forest patch and slope were associated with less mental health complaints, whereas composition related landscape metrics were not.

**Keywords:** physical activity; mental health; landscape metrics; hierarchical multiple regression

## 1. Introduction

There is abundant evidence that green space in urban neighborhoods is associated with physical activity [1–3]. Positive associations with increased levels of physical activity were reported for the amount of green space close to home [1], the distance to the nearest green space [4,5], the size of the nearest green space [5,6], and the presence of certain features [6]. Schipperijn et al. (2013) showed that urban green space can facilitate a wide range of free or low cost activities and the availability of urban green space is one of the environmental factors that is frequently linked to increased levels of physical activity [3].

It is well known that physical activity contributes to human health. In addition to fostering normal growth and development, it can reduce the risk of chronic diseases and conditions [7,8] such as depression [9], anxiety [10], and stress [11], and also can make people feel better, function better, and sleep better [12]. Evidence also shows that exposure to natural places can enhance psychophysiological recovery [13], and lead to positive mental health outcomes [14]. Contact with natural environments can similarly promote restoration from stress and mental fatigue [14,15].

The European Commission has devoted specific attention to nature-based solutions and has defined the concept within the range of ecosystem-based approaches [16]. Nature-based solutions aim to

help societies address a variety of environmental, social and economic challenges in sustainable ways. The concept builds on and supports other closely related concepts, such as ecosystem services, ecosystem-based adaptation/mitigation, and green and blue infrastructure. They all recognize the importance of nature and require a systemic approach to environmental change based on an understanding of the structure and functioning of ecosystems, including human actions and their consequences [17].

Forest ecosystem provide recreational services whether it be hiking, birdwatching, hunting, or taking in the scenic beauty and forests can generate recreational revenue through use- or entry-fees and ecotourism [18]. Agimass (2018) investigated the choice of forest sites for recreation [19] and Birch et al. (2014) describe that forests provide direct net income by charging visitors to access the picnic areas [20]. There is strong evidence that forests ecosystem can affect human health, well-being, and public health as well. Hunter et al. (2019) describe the relationship between duration of a nature experience, and changes in two physiological biomarkers of stress-salivary cortisol and alpha-amylase. They reported the efficiency of a nature pill per time expended was greatest between 20 and 30 min [21]. Focusing on forests, research reported that forest bathing, is associated with clinical therapeutic effects for the human physiological and psychological systems [22,23].

There are studies, on the other hand, that reported no association between green space and physical activity [24–26] or physical activity and health [27]. The research regarding physical activity as a mechanism explaining the relationship between green space and health has produced mixed results. Lachowycz and Jones (2011) identified 50 studies examining the relationship between green space and physical activity. Among them, only twenty studies (40%) reported a positive association between green space and physical activity. Two reported a negative association and 28 found no evidence for a relationship [28]. These mixed results were found at both a regional and national level.

Recent models of public health and environment reflect the fact that individual characteristics and preferences are active within the context of socio-economic, political, cultural and environmental factors that operate at different scales from the household and community to wider geographic levels [29]. Research since the 1990s using ecological models of behavior has increasingly emphasized the need to consider the physical environment more carefully in such studies [30,31]. This approach is of particular interest to those responsible for planning and designing the environment. A number of approaches to environment-behavior research have developed versions of this ecological model, reflecting similar understandings of the transactional nature of the relationship between person and place [32,33].

Within this context, public health policy has generally adopted a model of the relationship between environment and health that reflects Bronfenbrenner's human ecology theory where the individual is located within nested ecological systems [34]. Bronfenbrenner's work underlines how the individual can exert an influence over his or her environment and, at the same time, how the environment exerts an influence on the individual. Barton and Grant (2006) presented a conceptual model of the settlement as ecosystem in its context. The settlement ecosystem health map shows the relationship between health and the physical, social, and cultural environment (Figure 1). In this map, people are at the center of the built and natural environments, reflecting not only the focus on health, but also sustainable development activities. The human settlement is set within its natural environment and the global ecosystem on which it ultimately depends [29].

On a landscape scale, the conceptual models produced by Forman for the kinds of landscape patterns are necessary to support different groups of animal species—concepts such as the landscape mosaic of patch, corridor and matrix—which offer a basis for making planning decisions that minimize habitat loss [35,36]. During the past few decades, landscape pattern metrics have been employed for analyzing the composition and configuration of landscape structure and spatial pattern [37]. Composition refers to features associated with the variety and abundance of patch types within the landscape, but without considering the spatial character, placement, or location of patches within the mosaic. The principal metrics are number of categories, proportions, and diversity.

Spatial configuration refers to the spatial character and arrangement, position, or orientation of patches within the class or landscape [38]. Some aspects of configuration are measures of the spatial character of the patches themselves, even though the aggregation may be across patches at the class or landscape level.

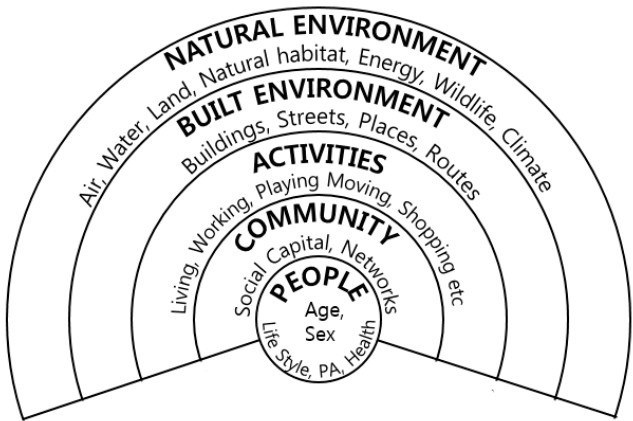

**Figure 1.** The relationship between health and environment (adapted from models on the determinants of health by Barton and Grant (2006)).

To date, it has been largely unclear how many green areas, and what size of natural elements are needed to achieve certain health benefits, and which type of green space will most benefit the health of local residents. The purpose of this study is to investigate the relationship between landscape structure, physical activity, and health and identify the factors of landscape structure and socioeconomic characteristics influencing adults' activity and health. Forest features such as size, shape, edge, number, slope, and elevation are discussed on the premise that they exert an influence on promoting physical activity and improving health.

## 2. Materials and Methods

### 2.1. Data

Data used in this study included the Korea Community Health Survey (KCHS), Land Use Land Cover (LULC), a digital elevation model (DEM), and geographic information system (GIS) files including administrative boundary (Figure 2). The KCHS data were collected from Korea Centers for Disease Control and Prevention in 2016. The centers conduct a nationwide survey to provide data for planning, monitoring, and evaluating community health promotion and disease prevention programs.

The survey was designed to establish a basis for implementing health services by producing community health statistics that establish and evaluate the community health care plan. It also tried to integrate evaluation indices for the health services of local governments by standardizing survey indices and procedures comparable among communities.

This survey was conducted by 253 community health centers, 35 community universities, and 1500 interviewers [39]. Among them, the data collected from 166 community health centers were used for this study. Seven metropolitan areas were excluded since the land cover conditions of those areas are not comparable for the calculation of landscape structure due to the severe urbanization.

LULC and DEM data were downloaded from the National Environmental Information Network System [40] and used to calculate landscape structure indices and topographic conditions respectively. LULC data consist of water, developed areas, barren, wetlands, forest, croplands, grasslands, and so forth. Among them, forest class was selected and exported in Grid file format in ArcGIS 10.1. The Grid file was converted to GeoTiff format for the calculation of landscape metrics in FRAGSTATS 4.2.1.

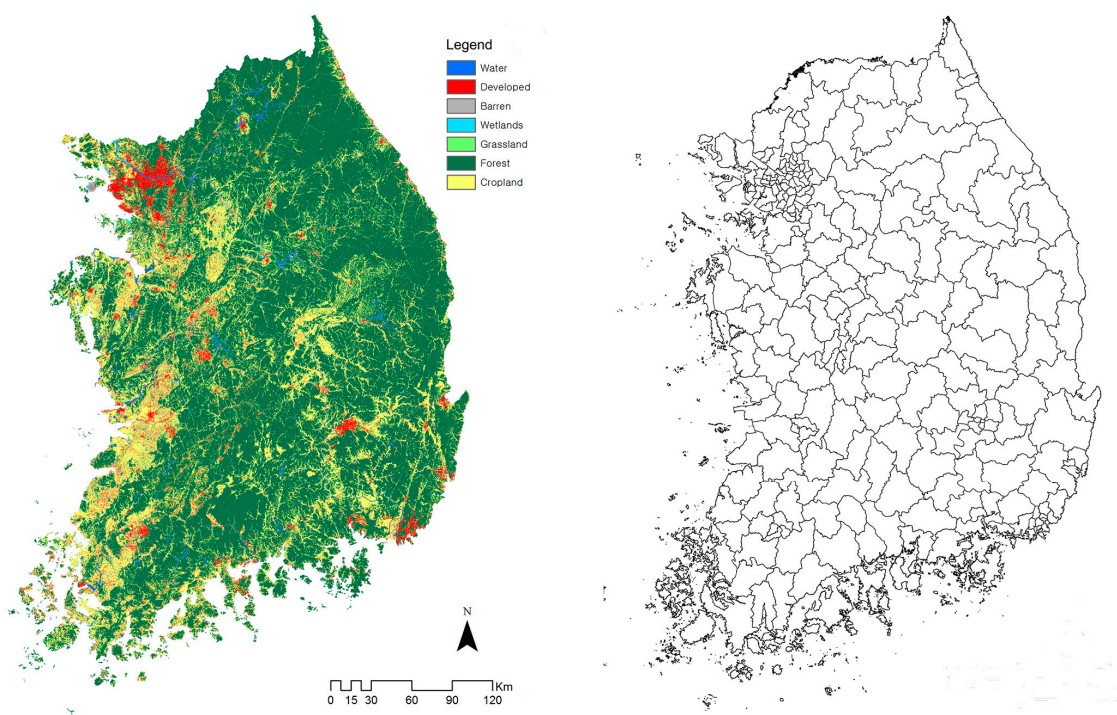

**Figure 2.** Land Use Land Cover map (**left**) and County boundary (**right**).

DEM data were used to calculate local elevation and slope. The DEM was clipped with the county boundary. The nearest-neighborhood resampling method was used to resample the created DEM in 30 m resolution. A spatial analysis was conducted to calculate elevation and slope.

*2.2. Study Population*

The target population was defined as adults aged over 19 years and who live in the jurisdiction of one community health center. The registered population data were obtained from the Ministry of Public Administration and Security. The data include gender, age, and population structure as well as the stratification of the surveyed population.

The sample size in each community health center was selected so that the main health index in each community health center has ±3% desired sampling error with a 95% confidence level. Nine hundred people on average were selected in each community health center and the total population was 148,754.

The KCHS collected information of many aspects of physical activity including running (jogging), tracking, biking, swimming, soccer, basketball, jump rope, squash, tennis, and other types of exercise. The metrics were level of physical activity in a typical day, weekly frequency of participation in vigorous physical activities for at least 10 min, and weekly frequency of participation in walking exercise for at least 10 min. The duration of these activities was included in this survey. Mental health status was measured by a question on subjective stress recognition and experience of depression.

The KCHS collected demographic and health-related information from participants. This included gender, age, education, smoking status, drinking, and oral and mental health. The metric of income had eight categories from low income of <$1000/month in a household to high income of >$5000/month.

*2.3. Landscape Structure*

The landscape structure was quantified with a number of landscape metrics. In this study, quantifying the landscape structure involved the use of statistics that described the landscape configuration and composition. Landscape metrics include Class Area (CA), Percent of Landscape (PLAND), Number of Patches (NP), Largest Patch Index (LPI), Edge Density (ED), and Mean Shape Index (MSI) (Table 1).

**Table 1.** Landscape metrics and description.

| Metrics | Formula | Description |
|---|---|---|
| Class Area (ha) | $CA = \sum_{j=1}^{n} a_{ij(\frac{1}{10,000})}$ | • CA equals the sum of the areas (m²) of all patches of the corresponding patch type |
| Percent of Landscape (%) | $PLAND = P_i \frac{\sum_{j=1}^{n} a_{ij}}{A}(100)$ | • %LAND equals the percentage the landscape comprised of the corresponding patch type. |
| Number of Patches | $NP = n_1$ | • NP equals the number of patches of the corresponding patch type (class) |
| Largest Patch Index | $LPI = \frac{\max\limits_{j=1}^{n} a_{ij}}{A}(100)$ | • LPI equals the percentage of the landscape comprised by the largest patch |
| Edge Density (m/ha) | $ED = \frac{\sum_{k=1}^{m'} e_{ik}}{A}(10,000)$ | • ED equals the sum of the lengths (m) of all edge segments involving the corresponding patch type, divided by the total landscape area (m²), multiplied by 10,000 |
| Mean Shape Index | $MSI = \frac{\sum_{j=1}^{n}\left(\frac{0.25 p_{ij}}{\sqrt{a_{ij}}}\right)}{ni}$ | • MSI equals the average shape index of patches of the corresponding patch type. When all patches are circular (vector) or square (raster) MSI equals 1. |
| Slope | | • Slope equals the average slope (%) of each county |
| Elevation | | • Elevation equals the average elevation of each county |

## 2.4. Statistical Analysis

The data were analyzed using SPSS 24. Descriptive characteristics of the study participants were tabulated. A hierarchical multiple regression analysis, controlling for socio-demographic characteristics, was used to investigate associations between landscape structure and physical activity and health. Figure 3 represents the research framework using landscape metrics, geographic information, and community health survey in this study.

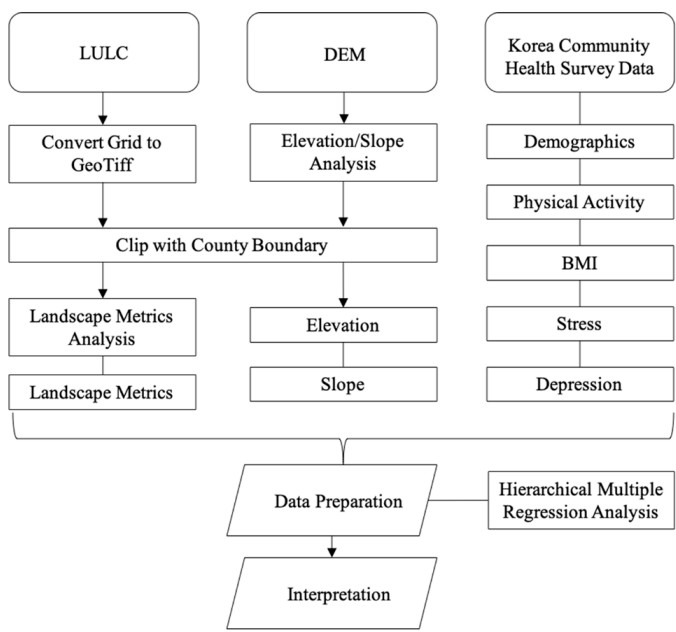

**Figure 3.** Framework of data used for geographic information and community health survey.

## 3. Results

We performed a hierarchical regression analysis to determine whether landscape structure influenced adults' physical activity and health beyond age, gender, education level, and household income. Eight predictors of landscape structure indices were used in the equation because of their statistically significant correlations with physical activity and health. Descriptive statistics and regression results are as follows.

### 3.1. Demographics

Of the 148,754 samples, 54.9% (81,665) were male and 45.1% (67,089) were female. According to the 2016 Korean Statistical Information Service data, the percentage of males and females in Korea is 50.12% and 49.88% respectively [41]. The average age of the participants was 54.57 years old and the oldest was 105. The highest participation age range was ages 50 to 59 (20.3%) and the lowest was over 80 (7.2%). The highest number of respondents for degree of education was secondary (30.8%) and the lowest degree of education held by the participants was graduate (1.5%). As for the monthly income, median income (ranging from $1000 to $4000) accounted for 50.3% of the participants and low income (<$1000/month in a household) occupied 24.1% (Table 2).

**Table 2.** Socioeconomic status.

| Variables | N | Percent |
|---|---|---|
| Age Goup (Years) | | |
| 19–29 | 13,992 | 9.4 |
| 30–39 | 18,376 | 12.4 |
| 40–49 | 25,591 | 17.2 |
| 50–59 | 30,143 | 20.3 |
| 60–69 | 26,279 | 17.7 |
| 70–79 | 23,611 | 15.9 |
| 80+ | 10,762 | 7.2 |
| Sex | | |
| male | 67,089 | 45.1 |
| female | 81,665 | 54.9 |
| Education | | |
| No school | 12,881 | 9.7 |
| Elementary | 30,821 | 23.1 |
| Subsecondary | 17,159 | 12.9 |
| Secondary | 40,974 | 30.8 |
| 2–3 years college | 15,497 | 11.7 |
| College | 13,644 | 10.3 |
| Graduate | 1967 | 1.5 |
| Income | | |
| 1 (lowest) | 14,611 | 9.9 |
| 2 | 20,985 | 14.2 |
| 3 | 25,233 | 17.1 |
| 4 | 26,628 | 18.1 |
| 5 | 22,318 | 15.1 |
| 6 | 15,240 | 10.3 |
| 7 | 8819 | 6.0 |
| 8 (highest) | 13,535 | 9.2 |

### 3.2. Physical Activity and Health

Exercise experts measure activity in metabolic equivalents (METs), which is defined as the energy it takes to sit quietly. For the average adult, this is about one calorie per every one kilogram of body

weight per hour. Vigorous-intensity physical activities (VPA) burn more than 6METs and include hiking, jogging, shoveling, bicycling fast, basketball, soccer games, and participating in a strenuous fitness class [42]. As for the frequency of VPA, 77.7% of respondents did not report vigorous physical activities. However, 3.6% respondents spent more than 10 min every day for physical activity (Table 3).

**Table 3.** Physical activity and health status.

| Variables | N | Percent |
|---|---|---|
| Physical Activity | | |
| 0 (lowest) | 115,333 | 77.7 |
| 1 | 6705 | 4.5 |
| 2 | 6330 | 4.3 |
| 3 | 6430 | 4.3 |
| 4 | 2664 | 1.8 |
| 5 | 4033 | 2.7 |
| 6 | 1606 | 1.1 |
| 7 (highest) | 5391 | 3.6 |
| Body Mass Index | | |
| underweight | 6890 | 4.9 |
| normal | 94,706 | 67.6 |
| overweight | 34,241 | 24.4 |
| obese | 4211 | 3.0 |
| Stress | | |
| 1 (highest) | 5343 | 3.6 |
| 2 | 31,054 | 20.9 |
| 3 | 76,346 | 51.3 |
| 4 (lowest) | 35,925 | 24.2 |
| Depression | | |
| 1 (highest) | 1645 | 1.1 |
| 2 | 18,810 | 12.6 |
| 3 (lowest) | 128,260 | 86.2 |

Regarding Body Mass Index (BMI), a measure of body fat based on height and weight that applies to adult men and women, 67.6% of respondents were normal, 24.4% of respondents were overweight, and 4.9% and 3% of respondents reported underweight and obese respectively. As for the health, while 24.2% of respondents did not feel stress at all, almost half of respondents felt stress a little, 20.9% felt stress considerably, and 3.6% felt strongly. In addition, 86.2% of respondents did not feel any depression at all, and 12.6% suffered from depression a little and 1.1% suffered from depression strongly.

*3.3. Landscape Metrics and Topography*

Table 4 indicates average values of landscape metrics and geographical characteristics. The average area of forest (CA) is 34,456.92 ha and the average proportion of the class (PLAND) is 58.88%. The average values of number of forest patches (NP), the largest forest percentage (LPI), edge density (ED), and mean shape of forest class (MSI) are 1576.14, 35.40%, 48.90 m/ha, and 1.23 respectively. The average slope of the study area is 19.93% and elevation is 294.67 m (Table 4).

**Table 4.** Descriptive Statistics.

| Metric Name | Acronym | Units | Mean | Std. Dev. | Range |
|---|---|---|---|---|---|
| Class Area | CA | ha | 34,456.92 | 30,894.44 | CA > 0, without limit |
| Percent of Landscape | PLAND | % | 58.88 | 20.77 | 0 < PLAND ≤ 100 |
| Number of Patches | NP | none | 1,576.14 | 1,237.77 | NP ≥ 1, without limit |
| Large Patch Index | LPI | % | 35.40 | 24.74 | 0 < LPI ≤ 100 |
| Edge Density | ED | m/ha | 48.90 | 15.67 | ED ≥ 0, without limit |
| Mean Shape Index | MSI | none | 1.23 | 0.062 | MSI ≥ 1, without limit |
| Slope | Slope | % | 19.93 | 8.94 | - |
| Elevation | Elevation | m | 294.67 | 190.95 | - |

### 3.4. Associations between Landscape Structure and Physical Activity and Health

Associations between landscape structure and overall physical activity, Body Mass Index, stress, and depression were observed in the hierarchical multiple regression analysis. Table 5 contains the standardized regression coefficients (*ßeta*), $R^2$, and change $R^2$ ($\Delta R^2$) in each health indicator. The Durbin-Watson statistics is a test for autocorrelation in the residuals from a statistical regression analysis. The Durbin-Watson statistic had a value from 1.655 to 1.911. This means that there is no autocorrelation detected in the sample.

**Table 5.** Hierarchical multiple regression analysis of health indicators by socioeconomic and landscape structure (N = 148,754).

| | Physical Activity (No. of Day) | | Body Mass Index | | Stress [1] | | Depression [2] | |
|---|---|---|---|---|---|---|---|---|
| | *ßeta* | | *ßeta* | | *ßeta* | | *ßeta* | |
| | Step 1 | Step 2 | Step 1 | Step 2 | Step 1 | Step 2 | Step 1 | Step 2 |
| Age | −0.100 ** | −0.107 ** | 0.013 ** | 0.011 ** | 0.242 ** | 0.235 ** | −0.043 ** | −0.047 ** |
| Gender | −0.117 ** | −0.118 ** | −0.164 ** | −0.163 ** | −0.033 ** | −0.033 ** | −0.088 ** | −0.087 ** |
| Education | −0.030 ** | −0.030 ** | −0.009 * | −0.010 * | 0.050 ** | 0.045 ** | 0.065 ** | 0.064 ** |
| Income | 0.036 ** | 0.036 ** | 0.039 ** | 0.042 ** | 0.003 | 0.012 ** | 0.098 ** | 0.104 ** |
| CA | | −0.029 ** | | 0.006 | | 0.002 | | 0.015 ** |
| PLAND | | 0.016 | | 0.009 | | −0.002 | | 0.000 |
| NP | | 0.025 ** | | 0.007 * | | 0.006 | | 0.005 |
| LPI | | 0.023 ** | | −0.012 * | | -0.005 | | −0.031 ** |
| ED | | −0.011 ** | | −0.009 * | | −0.020 ** | | −0.019 ** |
| MSI | | 0.044 ** | | 0.008 | | 0.019 ** | | 0.026 ** |
| Slope | | −0.015 | | −0.023 * | | 0.036 ** | | 0.068 ** |
| Elevation | | 0.010 | | 0.020 | | −0.001 | | −0.030 ** |
| $R^2$ | 0.025 | 0.027 | 0.027 | 0.028 | 0.044 | 0.046 | 0.043 | 0.045 |
| $\Delta R^2$ | - | 0.002 | - | 0.001 | | 0.002 | | 0.002 |
| *F* | 838.851 | 307.072 | 897.410 | 302.759 | 1532.116 | 300.721 | 1472.619 | 518.769 |
| Durbin-Watson | | 1.655 | | 1.911 | | 1.827 | | 1.787 |

* $p < 0.05$, ** $p < 0.01$, variable input: enter, (1) 1 = more stressful 4 = less stressful, (2) 1 = very depressed 3 = not depressed. CA: Class Area, PLAND: Percentage of Landscape, NP: Number of Patch, LPI: Largest Patch Index, ED: Edge Density, MSI: Mean Shape Index.

#### 3.4.1. Landscape Structure and Physical Activity

In Step 1, age, gender education, and income were forced into the equation, $R^2 = 0.025$, $F (4, 131852) = 838.851$, $p < 0.01$. The four predictors used in Step 1 accounted for 2.5% of the variance in physical activity. In Step 2, physical activity including socio-economic variables was plugged into the equation $R^2 = 0.027$, $F (12, 131844) = 307.072$, $p < 0.01$. Two point seven percent of the variance in physical activity was accounted for after Step 2. Comparison of the two steps also indicated that the change in $R^2$ was statistically significant ($\Delta R^2 = 0.002$, $p < 0.05$).

Significant associations between physical activity and landscape metrics in class area (CA), number of patches (NP), largest patch index (LPI), edge density (ED), and mean shape index (MSI)

were observed. Regression results showed that number of patches (NP), largest patch index (LPI), and mean shape index (MSI) were positively associated with physical activity while class area (CA) and edge density (ED) were negatively associated (Table 5).

### 3.4.2. Landscape Structure and BMI

The regression analysis showed that individual factors contributed significantly to the regression model, $R^2 = 0.027$, $F (4, 125330) = 897.410$, $p < 0.01$, and accounted for 2.7% of the variation in Body Mass Index in Step 1. In Step 2, Body Mass Index was plugged into the equation $R^2 = 0.028$, $F (12, 125322) = 302.759$, $p < 0.01$. Two-point eight percent of the variance in Body Mass Index was accounted for after Step 2. Comparison of the two steps also indicated that the change in $R^2$ was statistically significant ($\Delta R^2 = 0.001$, $p < 0.05$).

The findings also indicated that there were significant associations with number of patches (NP), largest patch index (LPI), edge density (ED), and slope. A regression analysis revealed that BMI was positively associated with number of patches (NP), whereas it was negatively associated with largest patch index (LPI), edge density (ED), and slope (Table 5).

### 3.4.3. Landscape Structure and Mental Health

In Step 1, age, gender education, and income were plugged into the equation $R^2 = 0.044$, $F (4, 131852) = 1532.116$, $p < 0.01$. Four-point four percent of the variance in stress was accounted for by the four predictors used in Step 1. In Step 2, stress was plugged into the equation, $R^2 = 0.046$, $F (12, 131844) = 531.833$, $p < 0.01$. Four-point six percent of the variance in stress was accounted for after Step 2. Comparison of the two steps also indicated that the change in $R^2$ was statistically significant ($\Delta R^2 = 0.002$, $p < 0.05$).

In the case of depression, the analysis revealed that individual factors contributed significantly to the regression model, $R^2 = 0.043$, $F (4, 131814) = 1472.619$, $p < 0.01$, and accounted for 4.3% of the variation in Step 1. In Step 2, depression was plugged into the equation $R^2 = 0.045$, $F (12, 131806) = 518.769$, $p < 0.01$ Four-point five percent of the variance in depression was accounted for after Step 2. Comparison of the two steps also indicated that the change in $R^2$ was statistically significant ($\Delta R^2 = 0.002$, $p < 0.05$).

Stress level was associated with edge density (ED), mean shape index (MSI), and slope significantly, whereas depression was associated with class area (CA), largest patch index (LPI), edge density (ED), mean shape index (MSI), slope, and elevation. Among these associations, regression results showed that stress was positively associated with MSI and slope while depression was associated with class area (CA), mean shape index (MSI), and slope (Table 5).

## 4. Discussion

Green space is increasingly considered as an important factor in relation to physical activity and mental health. Studies revealed that the amount of green space close to home and the distance and size of the nearest green space are positively associated with physical activity and health. This study examined the relationship between landscape structure, physical activity, and health with visually or psychologically sensible metrics focused on size, shape, number and edge of the forest patch.

Findings of this study show that the size, shape, and number of forest patches are positively associated with physical activity, whereas only the shape index is positively associated with mental health. Demographic characteristics in this study revealed that older people reported less physical activity and women demonstrated less physical activity than men. It was also found that those who have a higher education level reported less physical activity, whereas higher income was related to more physical activity (Table 6).

**Table 6.** The relationships among landscape structures, PA, and health indicators.

| | Model | PA | BMI | Stress | Depression |
|---|---|---|---|---|---|
| | Age | − | + | + | − |
| | Sex | − | − | − | − |
| 1 | Education | − | − | + | + |
| | Income | + | + | | + |
| | CA | − | | | + |
| | PLAND | | | | |
| | NP | + | + | | |
| | LPI | + | − | | − |
| 2 | ED | − | − | − | − |
| | MSI | + | | + | + |
| | Slope | | − | + | + |
| | Elevation | | | | − |

CA: Class Area, PLAND: Percentage of Landscape, NP: Number of Patch, LPI: Largest Patch Index, ED: Edge Density, MSI: Mean Shape Index.

## 4.1. Landscape Structure and Physical Activity

Hierarchical multiple regression analyses revealed that the largest forest patch and more irregular forest shape were positively related to physical activity. These results support the findings of previous studies that there are relationships between green spaces and physical activity [24,25,43]. The percentage of the largest forest patch was positively related among area related landscape metrics in this study. The total forest area, however, was negatively associated with physical activity and percentage of forest was not significantly related. It seems that only large forest area promotes respondents to perform frequent physical activity in the community.

The analysis also revealed that the number of forest patches was positively associated with physical activity. This result supports the findings of Kaczynski (2009). He examined how the number and total size of neighborhood parks within 1 km of participants' homes, as well as distance to the closest park, were related with physical activity. The study indicated that living near more parks and parkland showed more positive relationships with activity. A positive association was found with physical activity that may be more strongly linked to park or forest [1]. Based on this context, it appears that a greater number of forests is associated with better accessibility. However, landscape composition, which is an abundance of forest patches or proportion of green space, is less likely to be associated with physical activity.

A negative association was found with edge density (ED) and a positive association was found with mean shape index (MSI). Edge density is edge length on a per unit area basis, which facilitates comparison among landscapes of varying size. The forest edge effect results primarily from differences in wind and light intensity and quality reaching a forest patch, which alter microclimate and disturbance rates [44,45]. Edge index in a landscape is very critical information in the study of fragmentation and the total amount of edge is directly related to the degree of spatial heterogeneity. Spatial heterogeneity plays a critical role in determining ecological function. However, the results showed that it was negatively associated with physical activity (Table 6).

## 4.2. Landscape Structure and Health

The results showed that shape related landscape indices were associated with fewer mental health complaints and better body mass indices. Area related indices such as class area (CA), percentage of landscape (PLAND), number of patches (NP), and largest patch index (LPI) did not respectively show a significant relationship with stress, whereas mean shape index (MSI) and slope were positively associated with depression and edge density (ED) was associated negatively. With regard to depression, proportion and number of forest patches did not show a significant relationship with depression.

However, class area (CA), mean shape index (MSI), and slope were positively associated with depression while largest patch index (LPI), edge density (ED), and elevation were associated negatively (Table 6).

Several studies have shown that people who have access to green areas have less stress, and those who visit green areas more are less stressed than those who visit green areas less frequently [46,47]. Previous studies also indicated that human perception of landscapes is correlated with health and stress reduction [13], increased neighborhood satisfaction [48], and landscape structure. In this study, shape of the forest and slope positively influenced stress and depression. Since the shape and slope of forest are important elements of human perception of landscapes, the results support previous findings in relation to human perception and mental health. On the other hand, elevation negatively influenced to depression. As the elevation increases, people are more prone to depression. This result also supports previous findings [49,50] of the positive association between high altitude and the risk of depression, suicide.

## 5. Conclusions

This study identified the factors of landscape structure and socioeconomic characteristics influencing physical activity and health among adults who live in the jurisdiction of community health centers. A hierarchical multiple regression analysis was used to examine associations among these factors.

The study revealed that larger forest patches and more irregular forest shapes were associated with more physical activity, whereas total area of forest patches was negatively associated with physical activity. Landscape metrics, such as largest patch index (LPI), mean shape index (MSI), and class area (CA) were associated with physical activity with a significance level of 0.01.

Despite the importance of community-level contextual effects in health, few studies have focused on comprehensively understanding their aspects. The present study revealed that the shape of forest patches and slope are associated with fewer mental health complaints, whereas composition related landscape metrics, i.e., number or proportion of forest patches, were not associated with stress and depression. Mean shape index (MSI) and slope were positively associated with stress with a significance level of 0.01 while number of patch (NP) and percentage of landscape (PLAND) metrics were not.

This study offers a conceptual basis and a methodological framework for landscape ecological planning considerations for promoting physical activity and improving health. The results confirmed the usefulness of landscape metrics for estimating their relationships with physical activity and health, and deriving more quantitative evidence.

Understanding how elements of landscape structure influence people's physical activity and health is important for landscape architects, planners, and policy makers for planning and designing a healthy environment. For further studies, community capacity such as social capital as well as landscape structures should be considered as an independent variable for community-level contextual effects in public health.

**Author Contributions:** J.K. and D.-B.P. conceived the investigation, analyzed data and wrote the paper. J.K. acquired funding, realized literature review, analyzed GIS data sets, and performed images and tables production; J.I.S. provided revision of the written paper and analysis of data and was involved in project administration. All authors have read and agreed to the published version of the manuscript.

**Funding:** This work was supported by the research grant of the Kongju National University in 2019 (2019-0278-01).

**Conflicts of Interest:** The authors declare no conflict of interest.

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
