# Peer review of "Exploring the Relationship between Forest Structure and Health"

_forests, doi:10.3390/f11121264_

Round 1

Reviewer 1 Report

I have reviewed your responses to my questions and found them satisfactory for publication.

Reviewer 2 Report

Considering this being resubmitted article, I shall state that authors tried to cover my previous comments and suggestions. I agree with their improvements, however, I would like the discussion part to be improved in a more logic and concise way. I would highly suggest to put the table 5 in the “results” chapter.

Author Response

This manuscript is a resubmission of an earlier submission. The following is a list of the peer review reports and author responses from that submission.

Round 1

Reviewer 1 Report

Comments

Line

22        Data was collected in 2016, a lot has happened since then, any concerns that the data do

            not now represent a contemporary sample?

44        Recovery from what?

147      What is oral health?

190-194           Data suggest that the sample mostly consisted of relatively healthy individuals

254-256           Not quite, the data seem to suggest that certain forest structures often demand

                        higher levels of physical activity for participation. My read of the data don’t

                        suggest a causality between the two types of variables.

259-260           Not seeing the connection from the data of large forest sizes promoting physical

                        activity in the community. Perhaps I missed it.

267      Not seeing how number of forests is directly associated with issues of accessibility.

281      Edge is often associated with greater diversity of wildlife from which wildlife

            observation, using not highly physical, may be more prevalent.

287      Difficult at times to parse out differences in income and education, they are often highly

            related.

311-312           The sample did not report high levels of depression, without any explanation as to

                        why this might be the case leads this reviewer to view the results in this example

                        as spurious.

Reviewer 2 Report

I consider topic of the manuscript to be of high significance to readers as the potential positive impact of forests on the public health is recently perceived to be one of the most important ecosystems services provided by forests. However, to prove the hypothesis on existence of any relations between forest structure and health, it is necessary to come out with the scientifically sound paper which is not exactly the case of this manuscript. 

To follow the purpose of this study, I would recommend authors to consider and deal with the following suggestions:

  1. The introductory part is too general, describing only common ideas. I would appreciate if authors cover literature review on specific issues dealing with the recreational and/or therapeutic forest ecosystem services in connection to the public health.
  2. Considering methodological part, the landscape structure indicators are not really clear to me - what they stand for, why they were chosen, etc.
  3. Considering structure of the results, they need to be presented in much clearer form so that it is easy to catch the basic ideas of authors.
  4. The discussion part confronts the results with the authors of papers dealing with "green spaces" in general, not with "forest structures" in particular.
  5. The conclusions shall be thoroughly elaborated, following clear results supported by sound methodology approach.

Based on my suggestions, I would recommend authors to revise their manuscript accordingly.